# Mechanistic Insights into Salvigenin for Glucocorticoid-Induced Femoral Head Osteonecrosis: A Network Pharmacology and Experimental Study

**DOI:** 10.3390/biomedicines13030614

**Published:** 2025-03-03

**Authors:** Zhengjie Zhu, Yujian Zhong, Ruyuan He, Changheng Zhong, Junwen Chen, Hao Peng

**Affiliations:** 1Department of Orthopedics Surgery, Renmin Hospital of Wuhan University, Wuhan 430060, China; 2022283020081@whu.edu.cn (Z.Z.); zhongyujian@whu.edu.cn (Y.Z.); 18706827939@163.com (C.Z.); 2Department of Thoracic Surgery, Renmin Hospital of Wuhan University, Wuhan 430060, China; hry_whu@163.com

**Keywords:** network pharmacology, salvigenin, osteonecrosis of femoral head, osteoblast apoptosis, oxidative stress

## Abstract

**Background/Objectives:** Glucocorticoid-induced osteonecrosis of the femoral head (GIOFH) is a debilitating condition resulting from impaired bone metabolism and vascular disruption due to prolonged glucocorticoid use. This study aimed to explore the therapeutic potential of salvigenin, a flavonoid with antioxidative and estrogen-like properties, in alleviating GIOFH by modulating estrogen receptor alpha (*ESR1*) pathways. **Methods:** A network pharmacology approach was utilized to identify salvigenin’s potential targets and their association with GIOFH. Protein–protein interaction networks, along with Gene Ontology and KEGG pathway analyses, were conducted to clarify salvigenin’s multi-target mechanisms. Molecular docking and dynamics simulations assessed the interaction between salvigenin and *ESR1*. Experimental validation included in vitro assays on MG63 cells treated with dexamethasone (Dex) to mimic GIOFH, evaluating oxidative stress, apoptosis, osteogenic differentiation, and *ESR1* expression. **Results:** Network analysis identified *ESR1*, *NOS3*, and *MMP9* as key hub targets of salvigenin. Molecular docking and dynamics simulations confirmed stable binding of salvigenin to *ESR1*. Salvigenin significantly reduced Dex-induced oxidative stress and apoptosis in osteoblasts while restoring osteogenic differentiation and *ESR1* expression. Functional assays showed improved mineralized nodule formation, ALP activity, and mitochondrial integrity in salvigenin-treated cells. **Conclusions:** Salvigenin exhibits significant therapeutic potential in addressing GIOFH through *ESR1*-mediated pathways. These results offer a strong foundation for future translational studies and the development of salvigenin-based therapies for glucocorticoid-induced bone disorders.

## 1. Introduction

Glucocorticoid-induced osteonecrosis of the femoral head (GIOFH) is a severe condition marked by femoral head collapse resulting from disrupted bone metabolism and reduced blood supply, commonly caused by extended glucocorticoid exposure [1]. GIOFH accounts for a significant proportion of non-traumatic osteonecrosis cases worldwide, predominantly affecting middle-aged adults [2]. According to studies, the incidence of GIOFH has been reported to reach 9–40% in patients undergoing prolonged glucocorticoid therapy, particularly among those with conditions requiring long-term steroid treatment, such as systemic lupus erythematosus and rheumatoid arthritis [3]. Despite advances in glucocorticoid therapy, the associated risk of GIOFH remains a critical clinical challenge, often leading to severe disability and necessitating surgical interventions, such as total hip replacement [4]. The pathophysiology of GIOFH involves complex interactions between oxidative stress, apoptosis of osteocytes, vascular dysfunction, and disrupted bone remodeling [5,6,7,8]. Prolonged glucocorticoid exposure triggers excessive production of reactive oxygen species (ROS), disrupts mitochondrial integrity, and inhibits angiogenesis, collectively undermining the survival of osteoblasts and osteocytes [9]. Moreover, glucocorticoids alter cellular survival pathways, notably the PI3K/AKT and MAPK cascades, aggravating bone tissue injury [10]. These multifaceted mechanisms underscore the urgent need for novel therapeutic strategies targeting the molecular basis of GIOFH.

Emerging evidence indicates that estrogen receptor alpha (*ESR1*) is crucial for preserving bone homeostasis and promoting vascularization through its mediation of estrogen’s protective effects [11]. *ESR1* activation has been shown to counteract oxidative stress, promote osteogenic differentiation, and enhance angiogenesis, all of which are critical for preserving bone integrity in glucocorticoid-induced conditions [12]. Furthermore, *ESR1* is involved in the regulation of mitochondrial function and cellular antioxidant defenses, highlighting its potential as a therapeutic target for GIOFH [13].

Salvigenin, a naturally occurring flavonoid derived from medicinal plants such as Artemisia species (e.g., Artemisia annua), Salvia officinalis (sage), and Tanacetum parthenium (feverfew), has attracted significant attention for its diverse pharmacological properties, including anti-inflammatory, antioxidative, and estrogen-like activities [14]. Salvigenin has shown efficacy in regulating oxidative stress and preventing apoptosis across diverse disease models, such as neurodegenerative and cardiovascular disorders [15,16]. Its structural similarity to estrogen-related compounds suggests a potential role in activating *ESR1*, although its specific molecular interactions and therapeutic potential in GIOFH remain largely unexplored.

Network pharmacology, as a systems-level approach, provides a robust framework for elucidating the multi-target mechanisms of salvigenin by integrating chemical, biological, and pharmacological data [17]. This method facilitates the discovery of critical targets and pathways underlying disease mechanisms and therapeutic effects [18]. Integrating molecular docking with experimental validation, network pharmacology connects computational models to clinical applications, providing novel perspectives for multi-target drug development [19].

This study utilized network pharmacology to explore the therapeutic potential of salvigenin in modulating *ESR1* and its associated pathways to mitigate GIOFH. Protein–protein interaction (PPI) networks, Gene Ontology (GO) enrichment, and Kyoto Encyclopedia of Genes and Genomes (KEGG) pathway analyses were performed to develop a detailed interaction map. Additionally, molecular docking, alongside both in vitro and in vivo experiments, confirmed salvigenin’s efficacy in alleviating glucocorticoid-induced oxidative stress, maintaining osteocyte viability, and enhancing bone regeneration through *ESR1* activation. These results offer valuable insights into salvigenin’s role as a multi-target therapeutic agent for GIOFH and glucocorticoid-induced bone diseases.

## 2. Materials and Methods

### 2.1. Prediction of Potential Targets for Salvigenin

The potential targets of salvigenin were identified from multiple databases, including TCMSP (https://old.tcmsp-e.com/tcmsp.php, accessed on 1 December 2024), SuperPred (https://prediction.charite.de/, accessed on 1 December 2024), SwissTargetPrediction (http://www.swisstargetprediction.ch/, accessed on 1 December 2024), and BindingDB (https://www.bindingdb.org/rwd/bind/index.jsp, accessed on 1 December 2024). These databases utilize the structural features of the compound and previously known target data to provide a comprehensive range of potential targets. Meanwhile, targets related to osteonecrosis of the femoral head were retrieved from GeneCards (https://www.genecards.org/, accessed on 1 December 2024) and DisGeNET (https://www.disgenet.org/, accessed on 1 December 2024) using the keyword “Osteonecrosis of Femoral Head”. These sources integrate multidimensional data linking genes to diseases, ensuring the reliability and completeness of the retrieved target data. To standardize the analysis, gene symbols from all identified targets were unified using the UniProt database (https://www.uniprot.org/, accessed on 1 December 2024). Finally, common targets of salvigenin and osteonecrosis of femoral head were identified through intersection analysis, providing a foundation for investigating the molecular mechanisms of salvigenin in treatment.

### 2.2. Building the Protein Interaction Network and Identifying Key Targets

To investigate the shared targets of salvigenin and osteonecrosis of the femoral head, overlapping targets were analyzed using the STRING database (Version 12.0, https://cn.string-db.org/, accessed on 1 December 2024). The species was restricted to humans (*Homo sapiens*), and a medium confidence score threshold (>0.9) was applied to ensure high-quality PPI data. The network generated by STRING was exported and visualized with Cytoscape 3.9.1. In Cytoscape, the CytoHubba plugin evaluated the significance of each target within the PPI network. Among the available algorithms, the maximal clique centrality (MCC) method, known for its robustness in identifying hub nodes in complex networks, was applied. Targets were ranked based on their MCC scores, producing a prioritized list of key targets mediating salvigenin’s effects on GIOFH. This approach aids in identifying candidates for further experimental validation.

### 2.3. Differential Expression Analysis

The raw count data for GSE112101 were retrieved from the Gene Expression Omnibus (GEO) database. This dataset consists of transcriptomic sequencing of nine primary human cell types treated with glucocorticoids. For our analysis, we selected osteoblasts to investigate the impact of glucocorticoid treatment on osteoblast gene expression. Pre-processing and analysis were conducted using the DESeq2 package (version 1.38.0) in R. Initially, low-expression genes were filtered out by retaining those with at least 10 counts in over 50% of the samples. Library size normalization was applied using the median-of-ratios method implemented in DESeq2. Differential expression analysis was performed by fitting each gene to a negative binomial generalized linear model, and significance was evaluated through the Wald test. Genes with an adjusted *p*-value < 0.05 (Benjamini–Hochberg correction) and an absolute log2 fold change > 1 were classified as significantly differentially expressed. Data visualization included volcano plots and heatmaps, generated using the ggplot2 package. Subsequently, GO and KEGG pathway enrichment analyses were carried out on these significantly altered genes to determine their functional implications.

### 2.4. KEGG and GO Enrichment Analysis

KEGG pathway enrichment and GO functional annotation were performed using the R packages clusterProfiler and org.Hs.eg.db. The shared targets of salvigenin and GIOFH were first mapped from gene symbols to ENTREZ IDs through the bitr function in the clusterProfiler package, ensuring compatibility with downstream analyses. KEGG enrichment analysis utilized the enrichKEGG function with parameters set to *Homo sapiens* (hsa) and thresholds of *p* < 0.05 and q < 0.05. Similarly, GO annotation was conducted across the biological process (BP), molecular function (MF), and cellular component (CC) categories using the enrichGO function. Pathways and GO terms meeting significance criteria (*p* < 0.05) were visualized with the enrichplot and ggplot2 packages, offering insights into the biological functions and mechanisms underlying salvigenin’s therapeutic effects on GIOFH.

### 2.5. Gene Set Enrichment Analysis (GSEA)

GSEA was conducted using the clusterProfiler package (Version 4.8.3) in R, focusing on KEGG and GO databases. The input data consisted of a ranked gene list generated from the DESeq2 differential expression results, sorted by log2 fold change values in descending order. Default parameters were used for the analysis, and significantly enriched pathways were identified with an FDR threshold of <0.25.

### 2.6. Molecular Docking

The crystal structure of the protein target was retrieved from the PDB website (Protein Data Bank, https://www.rcsb.org/, accessed on 1 December 2024), and the three-dimensional structure of the ligand was downloaded from the PubChem database (https://pubchem.ncbi.nlm.nih.gov/, accessed on 1 December 2024). The protein structure was pre-processed using Pymol (version 2.5.0, https://pymol.org/, accessed on 1 December 2024), including the removal of water molecules, small ligands, and other heteroatoms. Hydrogen atoms and charges were added to the protein and ligand structures using AutoDock Tools (version 1.5.6), and the ligand’s rotatable bonds were identified and defined. The docking site was determined based on the binding pocket of the co-crystallized ligand or predicted through binding site prediction tools. The docking grid box was centered on the active site, with dimensions set to accommodate the ligand and ensure optimal sampling of the binding region. The docking simulations were analyzed using Autodock Vina (version 1.1.2), with exhaustiveness set to 8 to balance computational efficiency and accuracy. Post-docking analysis was performed to evaluate binding free energies (ΔG), and docking poses with the lowest ΔG values were considered as the most probable binding conformations. The docking results were visualized and dissected using Pymol and Discovery Studio Visualizer (version 2019, http://www.discoverystudio.net/, accessed on 1 December 2024). Additionally, theoretical inhibition constants (Ki and pKi) were calculated using the binding free energies based on the following equations:(1)$$K_{i}=e^{\frac{\Delta G}{RT}}$$(2)$$pK_{i}=-lg(K_{i})$$

Here, R represents the universal gas constant, and T represents the absolute temperature. These analyses provided quantitative insights into the binding affinity and potential inhibitory effects of the ligand on the target protein.

### 2.7. Molecular Dynamics Simulations

Molecular dynamics (MD) simulations were conducted using Gromacs 2022 to investigate interactions between the protein and the small-molecule ligand. The ligand was parameterized using the GAFF force field, while the AMBER14SB force field and TIP3P water model were applied to the protein and solvent to ensure the physicochemical accuracy of the system. During system construction, all water molecules, small ligands, and extraneous heteroatoms were removed, and periodic boundary conditions were employed to simulate biomolecular behavior in an infinite system. To maintain computational stability and precision, hydrogen bonds were constrained using the LINCS algorithm, with a time step of 2 femtoseconds. Electrostatic interactions were computed using the Particle Mesh Ewald (PME) method, with a cutoff distance set to 1.2 nm. Non-bonded interactions were updated every 10 steps. The temperature of the system was regulated at 298 K via the V-rescale coupling method, while the pressure was stabilized at 1 bar using the Berendsen coupling method, replicating physiological conditions. The simulation was divided into two stages: equilibrium simulations under NVT (constant volume and temperature) and NPT (constant pressure and temperature) ensembles ensured system stabilization, followed by 100 nanoseconds of production MD simulations. System snapshots were saved every 10 picoseconds to capture dynamic conformational changes. For trajectory analysis, VMD and PyMOL were used to visualize the stability of the ligand within the binding site of the protein. Binding free energies (ΔG) were calculated using the g_mmpbsa tool to quantify the binding affinity between the protein and ligand. Additionally, the contribution of key residues at the ligand-binding site could be extracted to provide insights into molecular interactions, offering guidance for further optimization of drug candidates.

### 2.8. Cell Culture of MG63 Cells

The MG63 human osteoblast-like cell line was sourced from Procell Life Science & Technology Co., Ltd. (Wuhan, China). The cells were maintained in Dulbecco’s Modified Eagle Medium (DMEM, Servicebio, Wuhan, China), supplemented with 10% fetal bovine serum (FBS, Procell, Wuhan, China) and 1% penicillin–streptomycin solution (Servicebio, China). Cultures were incubated at 37 °C in a humidified atmosphere containing 5% CO_2_. The culture medium was replaced every two days to maintain optimal cell health. When cells reached 80–90% confluency, they were subcultured using 0.25% trypsin-EDTA (Servicebio, China). The MG63 cell line was selected for this study because it is a widely used human osteoblast-like cell model in bone metabolism research, particularly in studies involving osteogenic differentiation, apoptosis, and oxidative stress. MG63 cells retain key osteoblastic characteristics and have been extensively utilized in in vitro models of bone-related diseases, including those induced by glucocorticoids. Their well-characterized biological properties and reproducibility make them an appropriate choice for investigating the potential protective effects of salvigenin on osteoblast function.

### 2.9. CCK-8

MG63 cells were utilized to assess the protective effects of salvigenin (MedChemExpress, Shanghai, China) against dexamethasone (Dex, MedChemExpress, Shanghai, China)-induced cytotoxicity, a cellular model for GIOFH. Cells were seeded at a density of 5 × 10^3^ cells per well in 96-well plates and cultured overnight in DMEM supplemented with 10% FBS and 1% penicillin–streptomycin to allow adherence. Subsequently, the cells were treated with 200 μM Dex for 24 h to induce cytotoxicity, followed by an additional 24-h treatment with varying concentrations of salvigenin (e.g., 0, 10, 20, 30, 40, 50 μM). Control groups received equivalent volumes of DMSO without Dex or salvigenin. Cell viability was evaluated using the Cell Counting Kit-8 (CCK-8, Beyotime, China) following the manufacturer’s instructions. Briefly, 10 μL of CCK-8 solution was added to each well, and the plates were incubated at 37 °C for 2 h. Absorbance was measured at 450 nm with a microplate reader (ThermoFisher Multiskan FC, Shanghai, China).

The percentage of cell viability was calculated using the formula:$$Cell\;Viability\left(\%\right)=\frac{{OD}_{sample}-{OD}_{blank}}{{OD}_{control}-{OD}_{blank}}\times 100$$

### 2.10. Apoptosis Detection

Osteoblast apoptosis was evaluated using an Annexin V-FITC/PI (Beyotime, Shanghai, China) dual-staining assay. MG63 cells were seeded into 24-well plates at a density of 3 × 10^4^ cells per well and treated with 200 μM Dex at 37 °C for 24 h to induce apoptosis. After treatment, cells were detached using trypsin, centrifuged at 1000 rpm for 5 min, and washed twice with PBS. The cell pellet was resuspended in 500 μL of 1× binding buffer at a concentration of 1–5 × 10^5^ cells/mL. To stain the cells, 5 μL of Annexin V-FITC and 10 μL of propidium iodide (PI) were added to the suspension. The mixture was incubated for 20 min at room temperature in the dark to prevent photobleaching. Following incubation, samples were immediately analyzed using a BD Influx flow cytometer, and apoptotic populations were quantified with CytExpert 2.0 software (Beckman Coulter, Lane Cove, NSW, Australia). Cells positive for Annexin V but negative for PI (Annexin V+/PI−) were categorized as early apoptotic cells, while those positive for both Annexin V and PI (Annexin V+/PI+) were identified as late apoptotic cells. The total apoptosis rate was determined by summing the percentages of early and late apoptotic cells. This method provided a precise quantification of Dex-induced apoptosis and allowed evaluation of salvigenin’s protective effects on osteoblast survival.

### 2.11. Alizarin Red S Staining

To investigate the effects of salvigenin on osteogenic differentiation in dexamethasone (Dex)-treated MG63 cells, Alizarin Red S staining was utilized to evaluate calcium deposition, a key indicator of matrix mineralization. MG63 cells were plated in 6-well plates and cultured in DMEM (Servicebio, China) supplemented with 10% FBS (Procell, China) and 1% penicillin–streptomycin (Servicebio, China). After 24 h of culture, cells were treated with 200 μM Dex to induce GIOFH, followed by salvigenin treatment. For osteogenic differentiation, the culture medium was replaced with DMEM containing 10 mM β-glycerophosphate (Solarbio, Beijing, China), 50 μg/mL ascorbic acid (Solarbio, Beijing, China), and 10% FBS. The medium was refreshed every 2–3 days during the treatment period. At the end of the experiment, cells were washed twice with phosphate-buffered saline (PBS) and fixed in 4% paraformaldehyde (Servicebio, China) at room temperature for 15 min. After fixation, cells were stained with 2% Alizarin Red S solution (pH 4.2, Sigma-Aldrich, Shanghai, China) for 20 min to detect calcium deposits. Excess stain was carefully removed by rinsing the cells with distilled water until the rinse solution became clear. Calcium deposition was then observed and imaged under a light microscope to qualitatively assess mineralization.

### 2.12. Alkaline Phosphatase (ALP) Staining

MG63 cells were seeded into 6-well plates and treated with 200 μM dexamethasone (Dex) to establish an GIOFH model, followed by salvigenin treatment in osteogenic differentiation medium. The differentiation medium consisted of DMEM (Servicebio, China) supplemented with 10 mM β-glycerophosphate, 50 μg/mL ascorbic acid, and 10% FBS (Procell, China), with medium refreshed every 2–3 days. On day 7, cells were fixed with 4% paraformaldehyde for 15 min at room temperature. Following fixation, an alkaline phosphatase (ALP) staining kit (e.g., Beyotime, China) was used according to the manufacturer’s protocol. After staining, cells were thoroughly rinsed with distilled water and examined under a light microscope (OLYMPUS IX71, Tokyo, Japan) to visualize ALP activity. This method provided a qualitative assessment of the early osteogenic differentiation effects of salvigenin.

### 2.13. Western Blot Analysis

Western blotting was used to analyze protein expression levels in MG63 osteoblast-like cells. Total protein was extracted using RIPA buffer with protease and phosphatase inhibitors (Servicebio, China). Protein concentrations were measured using a BCA assay kit (Beyotime, China), and equal amounts of protein were loaded onto a 15-lane SDS-PAGE gel for separation and transferred onto PVDF membranes. To ensure sufficient biological replicates while minimizing reagent waste, we adopted a strategic loading approach. Protein samples were loaded into designated lanes across the gel, and an additional set of samples was run on the opposite side of the same gel, ensuring each group had at least three independent biological replicates (*n* = 3) on a single membrane. Although only two representative bands per group are displayed, quantification was performed based on these replicates. Membranes were blocked with 5% non-fat milk in TBST and incubated with primary antibodies at 4 °C overnight, followed by HRP-conjugated secondary antibodies at room temperature for 1 h. After washing, protein bands were visualized using enhanced chemiluminescence (ECL, Biosharp, Hefei, China). GAPDH was used as an internal control to normalize protein loading, ensuring consistency across lanes. Since all replicates were analyzed on the same membrane, interblot control was not required. The sources and dilution ratios of all antibodies used in this study are provided in Table 1. Protein band intensities were quantified using ImageJ software (Version 1.54g).

### 2.14. Reactive Oxygen Species (ROS) Detection

Intracellular ROS levels were assessed using a commercially available ROS detection kit (Beyotime, China), following the manufacturer’s protocol. MG63 human osteoblast-like cells were cultured in 6-well plates and exposed to 200 μM dexamethasone (Dex) to induce oxidative stress, with subsequent salvigenin treatment. After the treatments, cells were incubated with 10 μM DCFH-DA in serum-free medium at 37 °C for 30 min in a dark environment to prevent probe degradation. Excess dye was carefully removed through multiple PBS washes to ensure minimal background interference. The fluorescence signal, indicative of intracellular ROS levels, was visualized and captured using a fluorescence microscope. Representative images were obtained for qualitative comparisons of ROS generation across experimental groups. This approach provided insights into the efficacy of salvigenin in mitigating Dex-induced oxidative stress in MG63 cells.

### 2.15. Mitochondrial Membrane Potential (ΔΨm) Detection Using JC-1 Assay

The mitochondrial membrane potential (ΔΨm) of MG63 cells was evaluated using the JC-1 assay kit (Beyotime, China) as per the manufacturer’s instructions. MG63 cells were cultured in 6-well plates and exposed to 200 μM dexamethasone (Dex) to induce mitochondrial dysfunction, followed by treatment with salvigenin. Post-treatment, cells were incubated with JC-1 staining solution at 37 °C for 20 min in a dark environment to ensure probe stability and prevent photobleaching. Excess dye was removed through washing steps with JC-1 staining buffer to eliminate non-specific fluorescence. Fluorescence signals, indicating changes in mitochondrial membrane potential, were immediately captured under a fluorescence microscope. This approach allowed qualitative visualization of mitochondrial health and provided insights into salvigenin’s protective effects against Dex-induced mitochondrial damage.

### 2.16. Immunofluorescence Staining

Immunofluorescence staining was employed to visualize the expression and localization of target proteins in MG63 human osteoblast-like cells. Cells were cultured on sterilized glass coverslips placed in 6-well plates. Once cells reached 70–80% confluence, they were treated with dexamethasone (Dex) and salvigenin according to the experimental protocol. Following treatment, cells were fixed with 4% paraformaldehyde for 15 min at room temperature to preserve cellular structures. To facilitate antibody penetration, cells were permeabilized using 0.1% Triton X-100 in PBS. Nonspecific binding sites were blocked with 5% bovine serum albumin (BSA, Servicebio, China) prepared in PBS. Primary antibodies specific to the target protein (ESR1, Proteintech, Wuhan, China) were applied overnight at 4 °C. After washing, fluorophore-conjugated secondary antibodies diluted in PBS containing 1% BSA were incubated for 1 h at room temperature in the dark to prevent photobleaching. Coverslips were mounted on glass slides using a fluorescence mounting medium containing DAPI (4′,6-diamidino-2-phenylindole, Servicebio, Wuhan China) to counterstain nuclei. Fluorescence signals were captured using an upright fluorescence microscope (Olympus BX53, Tokyo, Japan) equipped with appropriate excitation/emission filters. Representative images were obtained to evaluate the expression and subcellular localization of the target protein, providing insights into salvigenin’s effects on protein dynamics within MG63 cells.

### 2.17. Animal Model of GIOFH

Male Sprague–Dawley rats (8 weeks old, 200–250 g) were obtained from the Wuhan Institute of Biological Products (Wuhan, China) and housed in a controlled environment maintained at 22 ± 2 °C with a 12-h light/dark cycle. The rats were fed standard laboratory chow with free access to water and allowed to acclimate for one week prior to the experiment. To establish the GIOFH model, dexamethasone (Dex, 5 mg/kg; Sigma-Aldrich, St. Louis, MO, USA) was administered via intramuscular injection twice weekly over a four-week period. Control animals received equivalent volumes of saline injections. During the modeling process, rats were observed daily for signs of stress or adverse reactions, and their body weights were recorded weekly to monitor general health. At the conclusion of the experiment, animals were euthanized via CO_2_ asphyxiation, and femoral heads were harvested for histological and molecular analyses. Bone samples were fixed in 4% paraformaldehyde, decalcified using 10% EDTA (pH 7.4), and embedded in paraffin for sectioning. Histological assessment was performed on hematoxylin and eosin (H&E)-stained sections to evaluate trabecular bone structure and osteocyte viability.

### 2.18. Hematoxylin and Eosin (H&E) Staining

Femoral heads were immersed in 4% paraformaldehyde for fixation at room temperature for 48 h and then decalcified in 10% EDTA (pH 7.4) for a duration of 4 weeks. Decalcified samples were embedded in paraffin, and tissue sections with a thickness of 5 μm were prepared for staining. Hematoxylin and eosin (H&E) staining was conducted following established protocols to analyze trabecular bone integrity and osteocyte health. Tissue sections were subjected to deparaffinization in xylene, followed by gradual rehydration using a series of ethanol solutions. Hematoxylin staining was performed for 5 min to enhance nuclear visibility, after which sections were rinsed in running water, differentiated with 1% acid alcohol, and subsequently treated with ammonia water to intensify nuclear contrast. Cytoplasmic components were stained using eosin for 2 min, followed by sequential dehydration in graded ethanol and xylene. To finalize the preparation, sections were mounted with neutral resin and analyzed under a microscope (Olympus BX53, Japan), providing insights into trabecular bone microstructure and osteocyte morphology.

### 2.19. Immunohistochemistry (IHC)

Paraffin-embedded sections underwent deparaffinization in xylene and rehydration through a graded ethanol series before being rinsed with PBS. For antigen retrieval, the sections were immersed in citrate buffer (pH 6.0) and heated in a microwave oven for 10 min. After cooling to room temperature, sections were treated with 3% hydrogen peroxide for 10 min to eliminate endogenous peroxidase activity, followed by thorough washing with PBS. To minimize non-specific binding, sections were incubated with 5% BSA (Servicebio, China) in PBS for 30 min at room temperature. Primary antibodies specific to the target protein (e.g., ESR1, Proteintech, China) were applied overnight at 4 °C, diluted in PBS containing 1% BSA. The following day, sections were washed multiple times with PBS and incubated with horseradish peroxidase (HRP)-conjugated secondary antibodies (Proteintech, China) for 1 h at room temperature. The signal was visualized using diaminobenzidine (DAB) substrate solution (Beyotime, China) until a brown color developed, followed by counterstaining with hematoxylin for nuclear contrast. After washing, sections were dehydrated in a graded ethanol series, cleared in xylene, and mounted with neutral resin. Imaging was performed with an upright light microscope (Olympus BX53, Japan) to assess the localization and expression of the target proteins.

### 2.20. Statistical Analysis

All statistical evaluations were conducted using GraphPad Prism (version 9.0) or R software (version 4.2.0). The results are expressed as the mean ± standard deviation (SD) from a minimum of three independent experiments unless stated otherwise. Comparisons between two groups were performed using an unpaired Student’s *t*-test, while one-way analysis of variance (ANOVA) followed by Tukey’s post hoc test was employed for analyses involving multiple groups. For datasets that did not follow a normal distribution, non-parametric tests, such as the Mann–Whitney U test or Kruskal–Wallis test, were utilized. Statistical significance was defined as *p* < 0.05. Graphs were created with GraphPad Prism, and all experiments were conducted in triplicate to ensure reproducibility. Detailed statistical information, including *p*-values and effect sizes, is provided in the corresponding figure legends.

## 3. Results

### 3.1. Recognition of Salvigenin Targets for GIOFH Treatment

The molecular structure of salvigenin is depicted (Figure 1A). A total of 383 GIOFH-related targets were identified, among which 18 overlapped with the predicted targets of salvigenin (Figure 1B). These overlapping targets suggest that salvigenin has potential therapeutic relevance for GIOFH.

### 3.2. Network Analysis of Key Targets

The protein–protein interaction (PPI) network of the 18 intersecting targets was constructed utilizing the STRING database (Figure 1C). Significant interactions were observed among the targets, highlighting their functional importance in GIOFH pathogenesis. Further analysis of the core network using MCC scoring identified highly interconnected hub genes (Figure 1D). Genes, such as *NOS3*, *MMP9*, and *ESR1*, showed high MCC scores, suggesting their central roles in salvigenin’s potential therapeutic mechanisms (Figure 1E). These hub genes are likely critical mediators of the compound’s effects on GIOFH.

### 3.3. Functional and Pathway Enrichment Analysis

Pathway enrichment analysis indicated that salvigenin’s targets are significantly enriched in key pathways, including the thyroid hormone signaling pathway, estrogen signaling pathway, and fluid shear stress-related pathways, among others (Figure 1F). These pathways are closely associated with critical biological processes underlying GIOFH, such as oxidative stress regulation and bone metabolism. Gene Ontology (GO) enrichment analysis categorized these targets into biological processes, molecular functions and cellular components (Figure 1G). Key biological processes included responses to oxidative stress, regulation of collagen metabolism, and wound healing. Enriched molecular functions such as transcription regulator activity and serine hydrolase activity provide further insights into the multi-faceted mechanism of salvigenin in GIOFH treatment.

### 3.4. Transcriptomic and Pathway Insights into Osteoblast Dysfunction Induced by Glucocorticoid Treatment

RNA-seq analysis was conducted to investigate the transcriptomic alterations in osteoblasts following glucocorticoid treatment. Differential expression analysis recognized notable changes in gene expression, as shown in the volcano plot (Figure 2A). Enrichment analysis uncovered that these genes were involved in pathways, such as TNF signaling, apoptosis, and osteoblast differentiation, highlighting the impact of glucocorticoids on essential biological processes (Figure 2B). Gene Ontology (GO) analysis further identified significant enrichment in processes, like epithelial cell proliferation, oxidative stress regulation, and transcriptional activity, all of which are critical for maintaining osteoblast function (Figure 2C). Hierarchical clustering analysis revealed distinct expression patterns among the significantly altered genes, with co-regulated gene modules likely influenced by glucocorticoid treatment (Figure 2D). Notably, GSEA outcomings demonstrated that the ovarian steroidogenesis pathway was significantly activated (Figure 2F), whereas estrogen receptor-related modules, including G protein-coupled estrogen receptor activity and nuclear estrogen receptor activity, were markedly suppressed (Figure 2H,I). These findings are particularly significant given the pivotal role of *ESR1* (estrogen receptor 1) in osteoblast function and bone metabolism. Building on prior PPI network results, *ESR1* was recognized as a central hub gene with a high MCC score (Figure 1D). The suppression of *ESR1* under glucocorticoid treatment suggests that it may play a vital role in mediating the detrimental effects of glucocorticoids on osteoblast function. Additionally, glucocorticoid-induced processes, such as cellular stress response, apoptosis regulation, and extracellular matrix remodeling, were tightly linked to *ESR1*-regulated pathways (Figure 2E,G). Collectively, these findings underscore the critical role of *ESR1* as a promising therapeutic target for alleviating glucocorticoid-induced osteoblast dysfunction.

### 3.5. Molecular Docking and Dynamics Simulation of Salvigenin Binding to ESR1

To investigate the interaction between salvigenin and *ESR1*, molecular docking and molecular dynamics (MD) simulations were conducted. The docking results revealed that salvigenin binds within the ligand-binding domain of *ESR1*, forming key hydrogen bonds and hydrophobic interactions with residues, such as LEU347, THR347, LYS530, LEU525, and VAL534, as shown in the docking pose and 2D interaction diagram (Figure 3A). These interactions suggest a stable and specific binding mode for salvigenin to *ESR1*. MD simulations over a 100 ns trajectory confirmed the stability of the salvigenin–*ESR1* complex. The root mean square deviation (RMSD) analysis indicated that the complex achieved equilibrium early in the simulation, with minimal fluctuations compared to the free protein and ligand (Figure 3B). The radius of gyration (Rg) remained stable throughout, reflecting the structural compactness of the complex (Figure 3C). Additionally, root mean square fluctuation (RMSF) analysis identified dynamic flexibility in specific regions, highlighting the stability of the ligand-binding domain (Figure 3D). Protein–ligand interaction dynamics, including distance and solvent-accessible surface area (SASA), further validated the sustained binding of salvigenin to *ESR1* (Figure 3E,F). The binding free energy decomposition revealed favorable contributions from van der Waals and electrostatic interactions (Figure 3H), while key residues, such as GLU353, ARG394, and ASP351, were identified as major contributors to the binding energy (Figure 3I). Notably, hydrogen bond analysis showed consistent interactions between salvigenin and *ESR1* throughout the simulation, further supporting the stability of the complex (Figure 3J). These results demonstrate that salvigenin establishes a stable and specific interaction with *ESR1*, providing structural and energetic evidence for its potential regulatory role in *ESR1*-mediated pathways.

### 3.6. Salvigenin Protects Osteoblasts from Glucocorticoid-Induced Apoptosis Through ESR1 Modulation

To assess the protective effects of salvigenin on osteoblast viability, CCK-8 assays were performed. Glucocorticoid (Dex) treatment significantly reduced cell viability in a dose-dependent manner (Figure 4A), whereas salvigenin alone did not exhibit cytotoxic effects at the tested concentrations (Figure 4B). Co-treatment with salvigenin significantly restored cell viability in Dex-treated osteoblasts in a dose-dependent manner, suggesting its protective role (Figure 4C). Immunofluorescence analysis showed that Dex treatment dramatically downregulated *ESR1* expression in osteoblasts, while salvigenin co-treatment restored *ESR1* expression levels (Figure 4D). Western blot analysis further validated these results, showing that *ESR1* protein levels were significantly reduced by Dex but rescued by salvigenin treatment (Figure 4E,F). Additionally, salvigenin modulated the expression of apoptosis-related proteins: Bax (pro-apoptotic) levels were increased by Dex and reduced by salvigenin (Figure 4G), whereas Bcl-2 (anti-apoptotic) levels were decreased by Dex and upregulated by salvigenin (Figure 4H). Caspase-3 activation, a hallmark of apoptosis, was significantly induced by Dex and attenuated by salvigenin (Figure 4I). Flow cytometry analysis of apoptosis further supported these findings. Dex treatment led to a significant increase in apoptotic cell populations, while co-treatment with salvigenin markedly reduced apoptosis rates (Figure 4J,K). These results demonstrate that salvigenin protects osteoblasts from glucocorticoid-induced apoptosis, potentially through the restoration of *ESR1* expression and modulation of apoptosis-related signaling pathways.

### 3.7. Salvigenin Restores Osteogenic Differentiation Impaired by Glucocorticoids

To evaluate the effects of salvigenin on glucocorticoid-induced inhibition of osteogenic differentiation, we conducted a series of functional assays. Alizarin red staining revealed that Dex treatment significantly reduced mineralized nodule formation in osteoblasts, indicating impaired osteogenic capacity. Co-treatment with salvigenin markedly restored mineralized nodule formation, suggesting its potential to counteract glucocorticoid-induced osteogenic suppression (Figure 5A, upper panel). Similarly, alkaline phosphatase (ALP) staining showed decreased ALP activity in Dex-treated cells, which was partially rescued by salvigenin co-treatment (Figure 5A, lower panel). Immunofluorescence analysis demonstrated that the expression of osteogenic markers, such as osteopontin (OPN) and RUNX2 was significantly reduced in Dex-treated osteoblasts. Salvigenin co-treatment restored the expression of both OPN and RUNX2 to levels comparable to the control group, highlighting its ability to enhance osteogenic marker expression (Figure 5B,C). These results collectively suggest that salvigenin mitigates the deleterious effects of glucocorticoids on osteogenic differentiation, potentially through restoring osteogenic marker expression and mineralization capacity.

### 3.8. Salvigenin Suppresses Glucocorticoid-Induced Oxidative Stress in Osteoblasts

To investigate the effects of salvigenin on glucocorticoid-induced oxidative stress, we evaluated reactive oxygen species (ROS) levels and mitochondrial morphology in osteoblasts. DCFH-DA staining revealed a significant increase in intracellular ROS levels following Dex treatment, indicating heightened oxidative stress. Co-treatment with salvigenin markedly reduced ROS levels, suggesting its antioxidant effects (Figure 6A). Furthermore, JC-1 staining was used to assess mitochondrial membrane potential, as normal, polarized mitochondria exhibit red fluorescence, whereas depolarized mitochondria show green fluorescence. Dex treatment led to mitochondrial depolarization, as evidenced by an increase in green fluorescence and a decrease in red fluorescence. In contrast, co-treatment with salvigenin partially restored mitochondrial polarization, as indicated by an increased red-to-green fluorescence ratio (Figure 6B). These findings demonstrate that salvigenin alleviates glucocorticoid-induced oxidative stress in osteoblasts, potentially by reducing ROS levels and preserving mitochondrial function, which may contribute to its protective role in maintaining osteoblast viability and function.

### 3.9. Salvigenin Alleviates Glucocorticoid-Induced Femoral Head Necrosis

To assess the protective effects of salvigenin on glucocorticoid-induced femoral head necrosis, we performed histological and immunohistochemical analyses of the femoral head. Hematoxylin-eosin (H&E) staining revealed improved trabecular bone structure. Histological evaluation using H&E staining (Figure 7A) showed distinct differences among the experimental groups. In the control group, the femoral head exhibited well-preserved trabecular bone with dense osteocyte distribution within intact lacunae. However, in the Dex-treated group, there were significant pathological changes, including disorganized trabecular bone, increased empty lacunae, and reduced osteocyte density, indicative of severe osteonecrosis. Notably, co-treatment with salvigenin (Dex+salvigenin group) significantly mitigated these pathological changes, demonstrating improved trabecular structure, reduced empty lacunae, and enhanced osteocyte preservation. These results suggest that salvigenin effectively protects against glucocorticoid-induced femoral head necrosis. Immunohistochemical staining of *ESR1* was performed to evaluate the molecular mechanisms underlying the protective effects of salvigenin (Figure 7B). The control group showed strong *ESR1* expression, as indicated by dense brown staining in osteocytes and surrounding bone matrix. In the Dex-treated group, *ESR1* expression was markedly diminished, consistent with disrupted bone homeostasis and increased osteonecrosis. Importantly, salvigenin co-treatment significantly restored *ESR1* expression, with staining levels approaching those observed in the control group. These findings indicate that salvigenin enhances *ESR1* expression, which may contribute to its protective effects by promoting bone homeostasis and mitigating glucocorticoid-induced bone damage.

## 4. Discussion

This study explored the potential therapeutic role of salvigenin in glucocorticoid-induced osteonecrosis of the femoral head (GIOFH), focusing on its effects on oxidative stress, osteoblast viability, and osteogenic differentiation via the estrogen receptor alpha (*ESR1*)-mediated pathway. Through a combination of network pharmacology, molecular docking, experimental validation, and animal model studies, we demonstrated that salvigenin exhibits multi-target pharmacological activities, highlighting its promise as a therapeutic candidate for GIOFH.

Network pharmacology analysis revealed that salvigenin’s therapeutic potential is mediated by its interaction with key proteins, such as *ESR1*, NOS3, and MMP9, which were identified as hub targets within the protein–protein interaction (PPI) network. Functional enrichment analysis further suggested that these targets are involved in crucial biological processes, including oxidative stress regulation, bone metabolism, and vascularization, all of which are impaired in GIOFH [20,21]. Pathway analysis identified the estrogen signaling pathway as a major mechanism underlying salvigenin’s effects. These findings are consistent with previous studies highlighting the protective role of estrogen receptor activation in maintaining bone homeostasis and mitigating oxidative stress [22,23]. By targeting *ESR1*, salvigenin may restore disrupted signaling pathways in GIOFH, providing a mechanistic basis for its therapeutic effects.

Our in vitro experiments confirmed the protective role of salvigenin in dexamethasone (Dex)-induced osteoblast apoptosis. Dex treatment significantly increased apoptosis rates and altered the expression of apoptosis-related proteins, such as upregulation of Bax and downregulation of Bcl-2. Salvigenin co-treatment not only restored the balance of these proteins but also inhibited caspase-3 activation, suggesting its anti-apoptotic properties. Immunofluorescence and Western blot analyses further showed that salvigenin rescued the downregulation of *ESR1* caused by Dex, underscoring its role in restoring osteoblast viability through *ESR1*-mediated signaling pathways. These results align with prior studies demonstrating that *ESR1* activation can counteract Dex-induced apoptosis and promote cell survival [24].

The ability of salvigenin to restore osteogenic differentiation in Dex-treated osteoblasts was demonstrated through Alizarin Red S and ALP staining, both of which showed significant recovery of mineralized matrix formation and ALP activity. Additionally, salvigenin restored the expression of osteogenic markers, such as RUNX2 and OPN, which are critical regulators of bone formation. These findings suggest that salvigenin counteracts the inhibitory effects of Dex on osteogenesis, likely by modulating *ESR1* and associated pathways. The restoration of osteogenic differentiation is particularly important for bone regeneration in GIOFH, where impaired osteoblast function plays a central role in disease progression [25,26].

Oxidative stress is a key contributor to GIOFH, and Dex-induced ROS production disrupts mitochondrial function, leading to osteoblast apoptosis [27]. Salvigenin’s antioxidative properties were evident in its ability to significantly reduce ROS levels and restore mitochondrial integrity in Dex-treated cells. The preservation of mitochondrial membrane potential (ΔΨm) and morphology further supports salvigenin’s role in mitigating oxidative damage. These effects may be attributed to *ESR1* activation, which is known to enhance cellular antioxidant defenses and maintain mitochondrial function. By reducing oxidative stress, salvigenin not only protects osteoblasts but also creates a favorable environment for bone regeneration. However, while our study demonstrated salvigenin’s protective effects against GC-induced osteoblast injury, it remains unclear whether salvigenin affects the broader therapeutic functions of glucocorticoids, such as their anti-inflammatory and immunosuppressive effects. Future studies should investigate whether salvigenin modulates glucocorticoid receptor (GR) signaling in non-osteoblastic tissues, particularly in immune cells, to determine its potential impact on the systemic effects of glucocorticoid therapy.

In addition to in vitro findings, our animal model of GIOFH further validated the therapeutic effects of salvigenin. Dexamethasone-treated rats displayed typical features of GIOFH, including disrupted trabecular bone structure and increased empty lacunae. Salvigenin administration significantly mitigated these pathological changes, as evidenced by histological evaluation through H&E staining. Furthermore, immunohistochemical staining demonstrated that salvigenin restored *ESR1* expression in the femoral head, aligning with its in vitro effects and highlighting the consistency of its protective role in both cellular and physiological contexts.

The findings of this study provide a strong rationale for further investigation of salvigenin as a therapeutic agent for GIOFH. Its multi-target mechanisms, coupled with its ability to modulate *ESR1* and associated pathways, suggest that salvigenin could address multiple pathological aspects of GIOFH, including oxidative stress, apoptosis, and impaired osteogenesis. Moreover, the integration of network pharmacology, in vitro validation, and animal studies demonstrates the utility of systems-level approaches in identifying and characterizing novel therapeutic candidates. While these results are promising, further studies are needed to evaluate the pharmacokinetics, bioavailability, and long-term safety of salvigenin in vivo, as well as its potential synergistic effects with existing treatments.

Despite the comprehensive nature of this study, several limitations should be acknowledged. First, while the in vitro and animal model analyses provide valuable insights, the complex interactions within the human femoral head microenvironment require further investigation. Future studies should include clinical trials to validate the therapeutic effects of salvigenin in human GIOFH patients. Second, the specific molecular interactions between salvigenin and *ESR1* require deeper exploration using advanced techniques, such as cryo-electron microscopy. Thirdly, although salvigenin was shown to protect osteoblasts from GC-induced damage, its potential interactions with the immunosuppressive functions of glucocorticoids remain unexplored. Future investigations should assess whether salvigenin influences glucocorticoid receptor activation and downstream signaling pathways beyond osteoblasts, particularly in immune cells and other target tissues. Lastly, the potential off-target effects of salvigenin should be systematically assessed to ensure its safety and efficacy as a clinical candidate.

## 5. Conclusions

This study highlights the therapeutic potential of salvigenin in glucocorticoid-induced osteonecrosis of the femoral head (GIOFH) through its multi-target mechanisms, particularly its modulation of *ESR1*-mediated pathways. By alleviating oxidative stress, preventing osteoblast apoptosis, and promoting osteogenic differentiation, salvigenin presents a promising approach for mitigating the adverse effects of glucocorticoids on bone health. These findings hold significant clinical relevance for orthopedists and rheumatologists managing patients requiring long-term glucocorticoid therapy. Specifically, salvigenin-based interventions could delay or reduce the need for invasive surgical procedures (e.g., total hip replacement) by preserving femoral head integrity, thereby improving patients’ quality of life. For surgeons, this study provides a pharmacological rationale to explore salvigenin as a neoadjuvant therapy to enhance bone regeneration in early-stage GIOFH. Furthermore, the multi-target action of salvigenin suggests its potential as a complementary therapy to existing anti-resorptive agents (e.g., bisphosphonates) in preventing glucocorticoid-induced bone complications. These findings provide a strong foundation for future translational research focused on developing salvigenin-based therapies for GIOFH and other glucocorticoid-induced bone diseases.

## Figures and Tables

**Figure 1 biomedicines-13-00614-f001:**
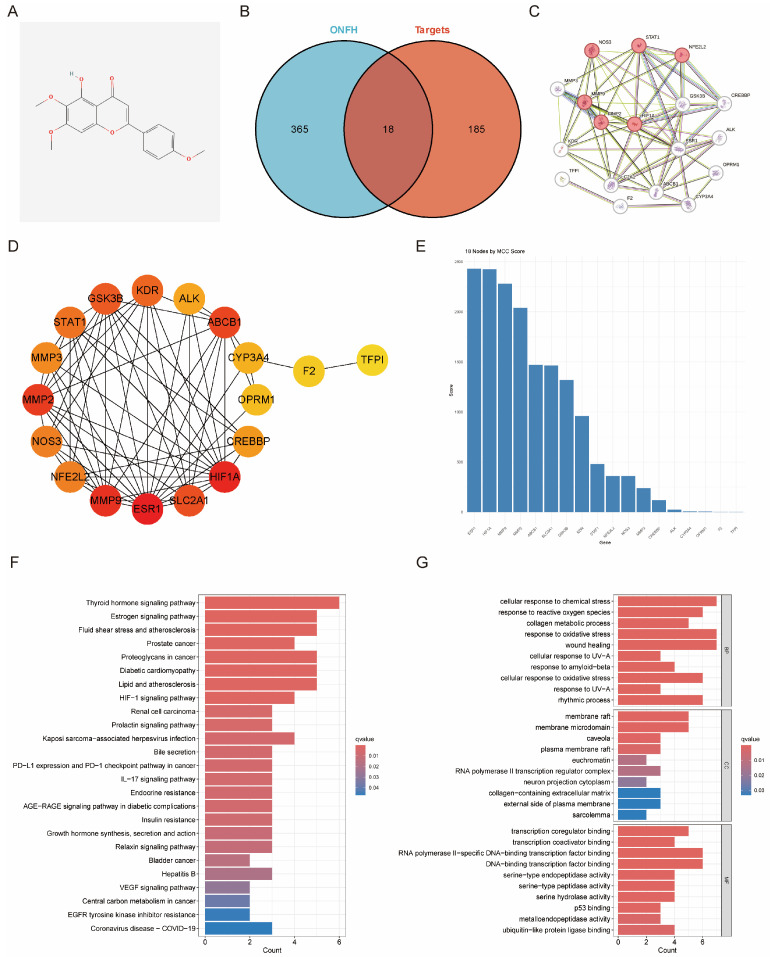
Target recognition and enrichment analysis of salvigenin for GIOFH. (**A**) Chemical structure of salvigenin. (**B**) Venn diagram showing the intersection between GIOFH-related and salvigenin-predicted targets, identifying 18 shared targets. (**C**) PPI network of the 18 overlapping targets constructed utilizing the STRING database, with edge confidence thresholds above 0.9. (**D**) Hub gene analysis of the PPI network using MCC scoring, highlighting the top-ranked genes (e.g., *ESR1*, *NOS3*, and *MMP9*) as critical targets of salvigenin. (**E**) MCC score ranking of the 18 overlapping genes, with *ESR1* identified as the top hub gene. (**F**) KEGG pathway enrichment analysis of the shared targets revealed significant enrichment in critical pathways, including the estrogen signaling pathway and the thyroid hormone signaling pathway. (**G**) Gene Ontology (GO) functional enrichment analysis of the shared targets highlights biological processes (top panel), cellular components (middle panel), and molecular functions (bottom panel) significantly linked to the therapeutic mechanisms of salvigenin.

**Figure 2 biomedicines-13-00614-f002:**
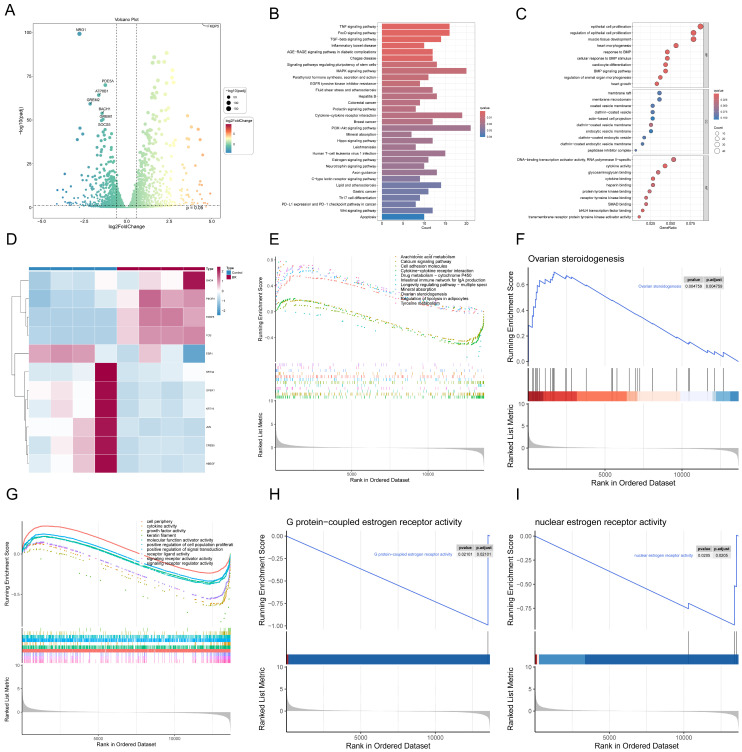
Transcriptomic and pathway insights into glucocorticoid-induced osteoblast dysfunction. (**A**) Volcano plot showing differentially expressed genes in osteoblasts following glucocorticoid treatment. (**B**) KEGG enrichment analysis of DEGs, highlighting key pathways, such as the TNF signaling pathway and osteoblast differentiation. (**C**) Gene Ontology (GO) enrichment analysis of DEGs categorized into biological processes, molecular functions, and cellular components. (**D**) Heatmap of hierarchical clustering analysis showing distinct expression patterns of DEGs across experimental conditions. (**E**) GSEA identifying significantly enriched pathways, including TNF signaling and oxidative stress-related pathways. (**F**) GSEA showing the upregulation of the ovarian steroidogenesis pathway in glucocorticoid-treated osteoblasts. (**G**) GSEA results indicating suppression of pathways related to G protein-coupled and nuclear estrogen receptor activities. (**H**) Enrichment plot for G protein-coupled estrogen receptor activity, showing marked suppression under glucocorticoid treatment. (**I**) Enrichment plot for nuclear estrogen receptor activity, demonstrating its significant downregulation in response to glucocorticoids.

**Figure 3 biomedicines-13-00614-f003:**
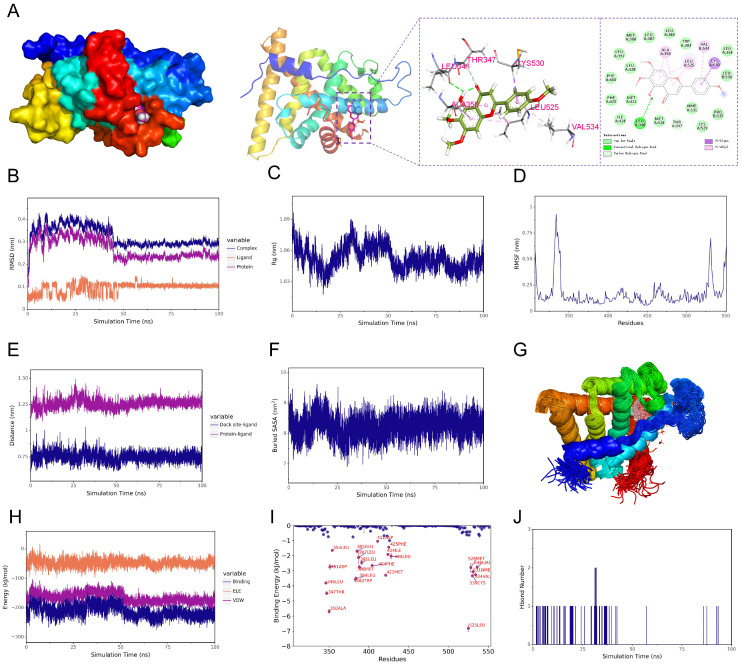
Molecular docking and molecular dynamics simulations of salvigenin binding to *ESR1*. (**A**) Docking results showing salvigenin binding within the ligand-binding domain of *ESR1*. Surface and ribbon models depict the protein structure, while the detailed 2D interaction map highlights key hydrogen bonds and hydrophobic interactions with residues LEU347, THR347, LYS530, LEU525, and VAL534. (**B**) Root mean square deviation (RMSD) plot showing structural stability of the protein–ligand complex, free ligand, and free protein during 100 ns molecular dynamics (MD) simulations. (**C**) Radius of gyration (Rg) analysis indicating the compactness of the *ESR1*–salvigenin complex throughout the simulation. (**D**) Root mean square fluctuation (RMSF) plot identifying flexibility variations across *ESR1* residues, highlighting stable regions in the ligand-binding domain. (**E**) Distance analysis showing stable docking interactions between the ligand and the active site during the MD simulation. (**F**) Solvent-accessible surface area (SASA) analysis of the protein–ligand complex, reflecting sustained interactions with the solvent environment. (**G**) Molecular dynamics trajectory showing the dynamic behavior of the *ESR1*–salvigenin complex, represented as a time-lapse ribbon model. (**H**) Binding energy decomposition over simulation time, indicating contributions from van der Waals, electrostatic, and binding energy terms. (**I**) Per-residue binding energy analysis highlighting key residues, such as GLU353 and ARG394, contributing to salvigenin binding. (**J**) Hydrogen bond occupancy histogram showing the frequency and stability of hydrogen bonds between salvigenin and *ESR1* throughout the simulation.

**Figure 4 biomedicines-13-00614-f004:**
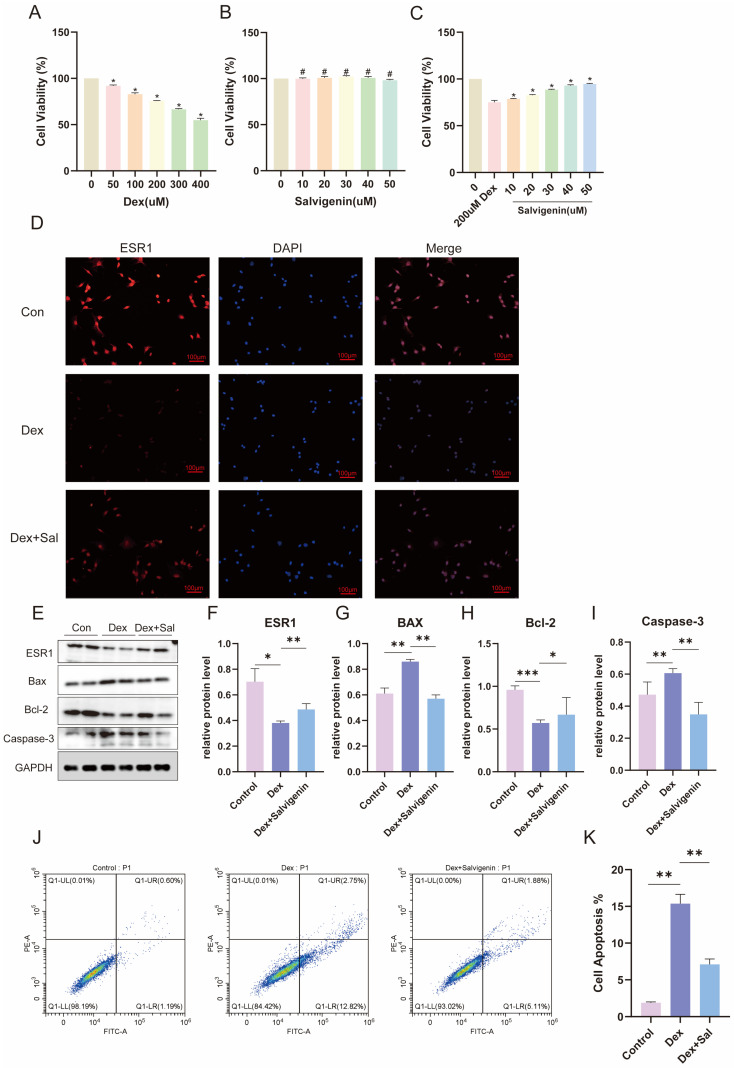
Salvigenin protects osteoblasts from glucocorticoid-induced apoptosis and restores *ESR1* expression. (**A**–**C**) Cell viability assays using CCK-8 (n = 3). (**A**) Dose-dependent cytotoxicity of dexamethasone (Dex) in MG63 cells. (**B**) Dose-response effects of salvigenin on cell viability, showing no cytotoxic effects up to 50 μM. (**C**) Salvigenin protects against Dex-induced cytotoxicity, restoring cell viability in a dose-dependent manner. * *p* < 0.05 compared to control; # *p* < 0.05 compared to Dex. (**D**) Immunofluorescence analysis of *ESR1* expression (red) in MG63 cells under different treatments: control, Dex, and Dex+salvigenin. Nuclei are stained with DAPI (blue). Merged images highlight the restoration of *ESR1* expression with salvigenin treatment. (**E**–**I**) Western blot analysis of apoptosis-related proteins and *ESR1*. (**E**) Representative protein bands for *ESR1*, Bax, Bcl-2, and Caspase-3, with GAPDH as the loading control. Quantification of (**F**) *ESR1*, (**G**) Bax, (**H**) Bcl-2, and (**I**) Caspase-3 protein levels, showing that salvigenin reverses Dex-induced apoptotic signaling (n = 3). * *p* < 0.05, ** *p* < 0.01, *** *p* < 0.001. (**J**,**K**) Flow cytometry analysis of apoptosis using Annexin V-FITC/PI staining. (**J**) Representative dot plots showing early and late apoptotic cell populations under control, Dex, and Dex+salvigenin treatments. (**K**) Quantification of total apoptosis rates, with salvigenin significantly reducing Dex-induced apoptosis (n = 3). ** *p* < 0.01.

**Figure 5 biomedicines-13-00614-f005:**
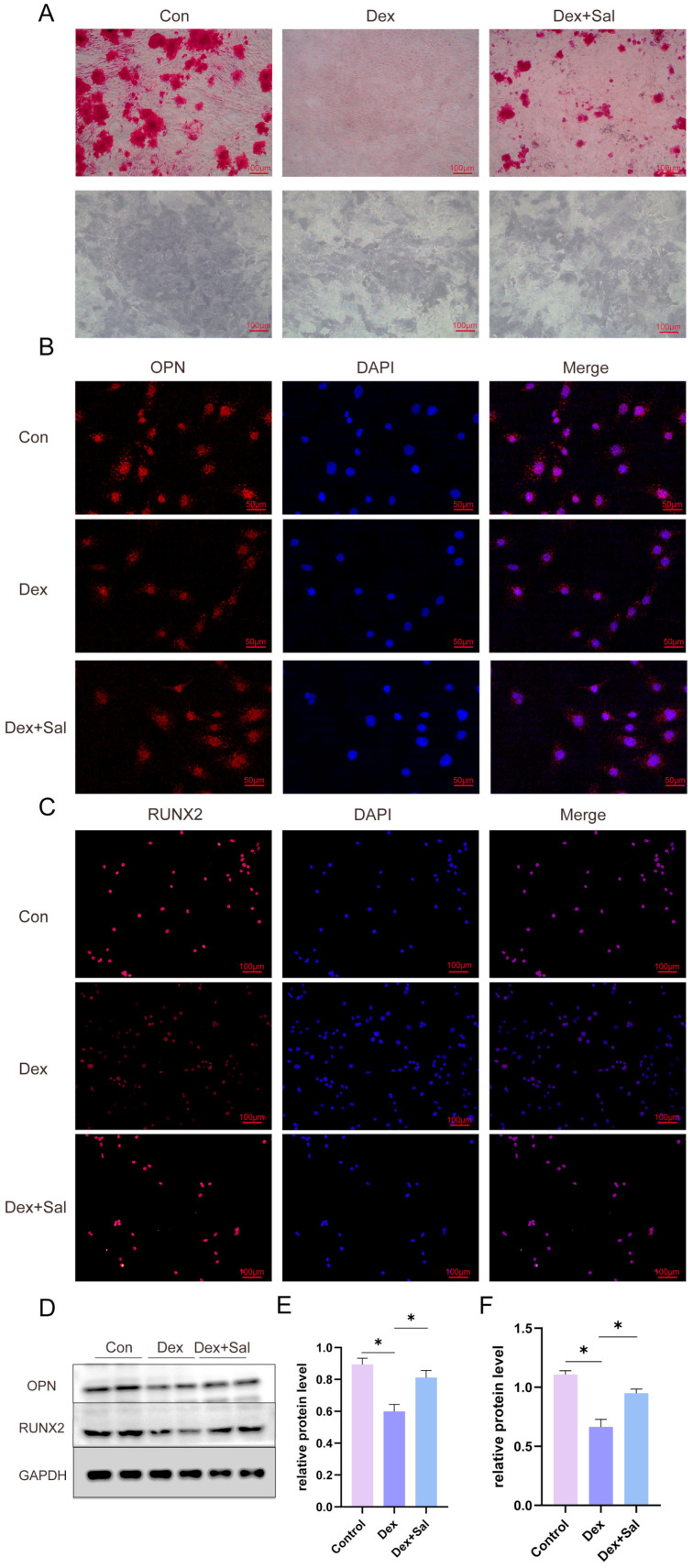
Salvigenin restores osteogenic differentiation impaired by glucocorticoids. (**A**) Alizarin Red S and ALP staining showing mineralized nodule formation (top panel) and alkaline phosphatase activity (bottom panel) in MG63 cells under control, Dex-treated, and Dex+salvigenin conditions. Salvigenin restores osteogenic capacity reduced by Dex. (**B**,**C**) Immunofluorescence analysis of osteogenic markers. (**B**) Expression of osteopontin (OPN, red) and (**C**) runt-related transcription factor 2 (RUNX2, red) in MG63 cells treated with Dex and salvigenin. Nuclei were counterstained with DAPI (blue). Merged images demonstrate salvigenin rescues osteogenic marker expression. (**D**–**F**) Western blot analysis of OPN and RUNX2. (**D**) Representative protein bands for OPN, RUNX2, and GAPDH (loading control). (**E**,**F**) Quantification of OPN (**E**) and RUNX2 (**F**) protein levels, showing significant restoration with salvigenin treatment (n = 3). * *p* < 0.05.

**Figure 6 biomedicines-13-00614-f006:**
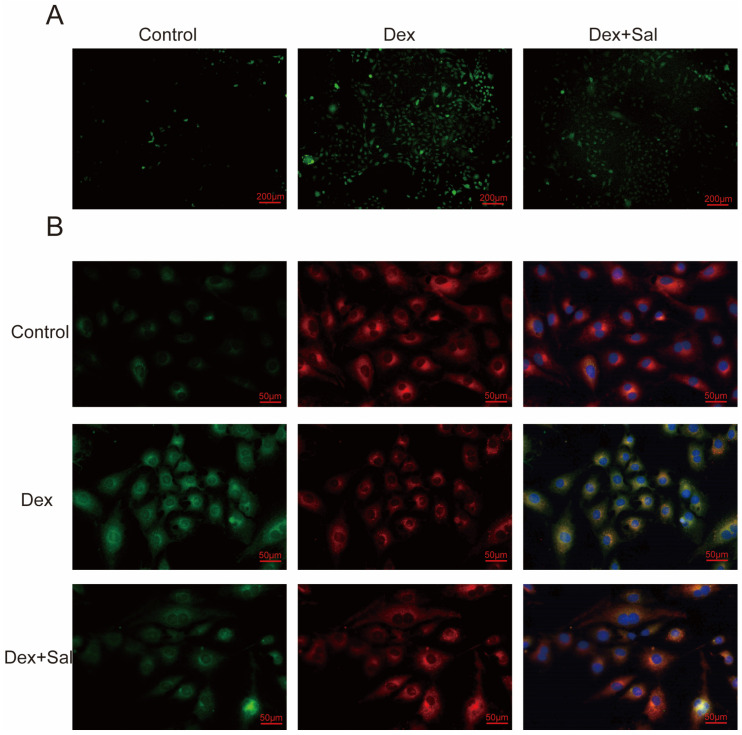
Salvigenin suppresses glucocorticoid-induced oxidative stress and preserves mitochondrial integrity in osteoblasts. (**A**) Intracellular reactive oxygen species (ROS) levels detected using DCFH-DA staining. Dex treatment significantly increases ROS levels, while co-treatment with salvigenin reduces oxidative stress. (**B**) Mitochondrial membrane potential assessed via JC-1 staining. Normal, polarized mitochondria exhibit red fluorescence, whereas depolarized mitochondria show green fluorescence. Dex treatment induces mitochondrial depolarization, increasing green fluorescence. Salvigenin co-treatment restores mitochondrial polarization, as indicated by an increased red-to-green fluorescence ratio.

**Figure 7 biomedicines-13-00614-f007:**
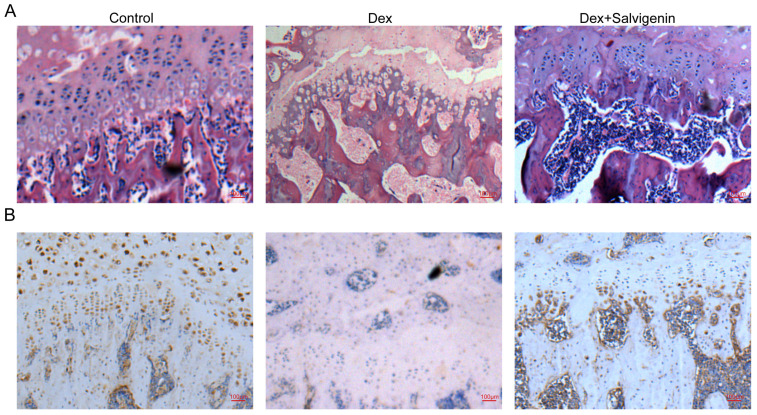
Salvigenin alleviates glucocorticoid-induced femoral head necrosis in vivo. (**A**) Hematoxylin and eosin (H&E) staining of femoral head sections showing trabecular bone structure and osteocyte distribution. The control group exhibits well-preserved trabecular bone with intact osteocytes, whereas the Dex-treated group shows disorganized trabeculae, increased empty lacunae, and reduced osteocyte density. Salvigenin co-treatment (Dex+salvigenin) significantly restores trabecular bone integrity and osteocyte preservation. (**B**) Immunohistochemical staining for *ESR1* in femoral head sections. The control group shows strong *ESR1* expression (brown staining), which is significantly reduced in the Dex-treated group. Salvigenin co-treatment restores *ESR1* expression levels, indicating its role in mitigating glucocorticoid-induced bone damage.

**Table 1 biomedicines-13-00614-t001:** The source and dilution concentrations of primary antibodies.

	Source	Dilutions
ESR1	Proteintech 21244-1-AP	1:1000
Bax	Proteintech 50599-2-Ig	1:8000
Bcl2	Proteintech 26593-1-AP	1:2000
Caspase3	Proteintech 25128-1-AP	1:1000
OPN	Proteintech 30200-1-AP	1:2000
RUNX2	Proteintech 20700-1-AP	1:1000
GAPDH	Proteintech 10494-1-AP	1:10,000

## Data Availability

Data are available in the public databases mentioned in the manuscript.

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
