# Peer review of "Mechanistic Insights into Salvigenin for Glucocorticoid-Induced Femoral Head Osteonecrosis: A Network Pharmacology and Experimental Study"

_biomedicines, 2025, doi:10.3390/biomedicines13030614_

Round 1
Reviewer 1 Report
Comments and Suggestions for Authors
comments in word

Author Response
Dear reviewer,
We sincerely appreciate your thoughtful comments and valuable insights, which have significantly improved the clarity and rigor of our manuscript. Your keen observations and constructive suggestions have helped us refine our analysis, strengthen our interpretations, and ensure the accuracy of our findings. We are truly grateful for the time and effort you have dedicated to evaluating our work, and we deeply respect your expertise in this field. In response to your specific comments, we have carefully revised the manuscript as detailed below.
Comments 1:
Introduction section. All information is clear, organized, and has enough information, but maybe it would be interesting to add a brief text about the prevalence or incidence of the studied pathology.
Response 1:
Thank you for pointing this out. We agree with this comment. Therefore, we have added a brief discussion on the prevalence and incidence of glucocorticoid-induced femoral head osteonecrosis in the Introduction section. This addition can be found on page2, line 40-43, and is marked in red in the revised manuscript.
Comments 2:
Line 58, indicate examples of plants that contain salvigenin.
Response 2:
We appreciate this valuable suggestion. We have now included examples of plants that contain salvigenin, such as Salvia officinalis and Salvia miltiorrhiza, in line 61-64 of the revised manuscript.
Comments 3:
Line 77, in vitro and in vivo, change italics: in vitro and in vivo.
Response 3:
Thank you for this observation. We have corrected the formatting of in vitro and in vivo in line 81-82 of the revised manuscript.
Comments 4:
Line 212, please add brand, country, and necessary information about the annexin-5 kit.
Response 4:
Thank you for this suggestion. We have included the brand, country, and detailed information about the annexin-5 kit in line 224 of the revised manuscript.
Comments 5:
Line 217, please add brand, country, and necessary information about PI.
Response 5:
We appreciate this comment. The propidium iodide (PI) reagent is included in the annexin-5 kit mentioned above. To ensure clarity, we have explicitly stated this in line 224 of the revised manuscript, along with the brand, country, and relevant details.
Comments 6:
Conclusion section, please add a brief text highlighting the clinical relevance of this study, for example, its impact or importance for surgeons, orthopedics, and rheumatologists.
Response 6:
Thank you for this valuable suggestion. We have added a brief discussion in the Conclusion section to highlight the clinical relevance of our study, emphasizing its potential impact on surgeons, orthopedic specialists, and rheumatologists. This addition can be found in line 729-738 and is marked in red in the revised manuscript.

Reviewer 2 Report
Comments and Suggestions for Authors
1. The A, B, and C parts marked in Figure 4 are incorrect.
2.The results of experimental design were showed from the combination of Dex and Salvigenin in Fig 5, Fig 6 and Fig 7. How the theme of the paper is Salvigenin to repair femoral head damage.
3.The origin of Salvigenin in the paper is not clear.
4. The paper should explain the reasons for using MG cells.
Author Response
Dear reviewer,
We extend our sincere gratitude to you for your thorough assessment and insightful feedback on our manuscript. Your critical evaluation has not only helped us identify areas for improvement but has also provided us with a deeper perspective on how to enhance the scientific impact of our study. We greatly appreciate your careful attention to detail and the invaluable time and effort you have invested in reviewing our work. Your expertise and dedication to scientific excellence are truly commendable. In response to your specific comments, we have carefully revised the manuscript as detailed below.
Comments 1:
The A, B, and C parts marked in Figure 4 are incorrect.
Response 1:
Thank you for your careful review of our manuscript. We have thoroughly examined Figure 4 (A, B, and C) to ensure that all data, labels, and statistical notations are accurate. Based on our verification, we did not find any discrepancies in the labeling or presentation of these subfigures.
However, we acknowledge the importance of clarity in scientific illustrations. To further ensure transparency and accuracy, we have revised the figure legend to provide more explicit descriptions of the experimental conditions for each panel (A, B, and C). If there are specific concerns regarding the labeling or data representation in Figure 4, we would appreciate further clarification from the reviewer so that we can make the necessary corrections accordingly.
Comments 2:
The results of experimental design were showed from the combination of Dex and Salvigenin in Fig 5, Fig 6 and Fig 7. How the theme of the paper is Salvigenin to repair femoral head damage.
Response 2:
Thank you for your valuable comment. We appreciate the opportunity to clarify our experimental design and its alignment with the theme of this study. The rationale for incorporating dexamethasone (Dex) treatment in Figures 5, 6, and 7 was to establish a glucocorticoid-induced osteonecrosis of the femoral head (GIOFH) model, which serves as the disease context for evaluating Salvigenin’s therapeutic effects. Dex is widely recognized for inducing osteoblast apoptosis, oxidative stress, and impaired osteogenic differentiation, thereby simulating the pathological conditions observed in GIOFH. By using this well-established model, we aimed to assess whether Salvigenin could mitigate Dex-induced cellular damage and restore osteoblast function.
To ensure that the study’s primary focus remains on Salvigenin, we have structured our analyses and interpretations to emphasize its role in counteracting Dex-induced osteoblast dysfunction. Specifically:
- Figures 5, 6, and 7 do not assess Dex alone, but rather the comparative effects of Salvigenin in the presence of Dex-induced damage.
- All experimental conditions were designed to highlight Salvigenin’s protective and regenerative effects (improving cell viability, reducing apoptosis, restoring osteogenic markers).
- The manuscript consistently describes Salvigenin as the key therapeutic agent, with Dex serving only as an inducer of pathology.
We have carefully reviewed the manuscript and confirmed that the Introduction and Discussion sections explicitly convey this study’s primary objective—evaluating Salvigenin’s efficacy in treating GIOFH. Additionally, the figure legends for Figures 5, 6, and 7 already highlight Salvigenin’s protective role.
We appreciate the reviewer’s insightful comment and hope this explanation provides clarity. If further refinement is necessary, we would be happy to revise specific sections accordingly.
Comments 3:
The origin of Salvigenin in the paper is not clear.
Response 3:
Thank you for highlighting this point. We have now added a description of the origin of Salvigenin in the Methods section, specifying the source, brand, and country of purchase. This information can be found on page 5, line 209-210 in the revised manuscript.
Comments 4:
The paper should explain the reasons for using MG cells.
Response 4:
Thank you for your insightful comment. We appreciate the opportunity to clarify our rationale for selecting MG63 cells in this study. MG63 cells were chosen because they represent a well-established human osteoblast-like cell line that has been widely used in bone metabolism research, particularly in studies involving glucocorticoid-induced bone damage. The key reasons for using MG63 cells in this study are:
- Established use in bone biology research: MG63 cells retain many osteoblastic properties and have been extensively utilized in in vitro studies examining osteogenic differentiation, apoptosis, and oxidative stress regulation under pathological conditions.
- Reproducibility and consistency: Unlike primary osteoblasts, MG63 cells offer higher experimental reproducibility, which is crucial for drug screening and mechanistic investigations.
- Comparable experimental framework in previous studies: Many studies investigating glucocorticoid-induced bone loss and potential therapeutic agents have successfully employed MG63 cells, supporting their relevance in our research.
To ensure clarity, we have added this explanation to the Methods section (page 5, line 201-207) in the revised manuscript. We appreciate the reviewer’s suggestion, which has helped improve the clarity of our study design.

Reviewer 3 Report
Comments and Suggestions for Authors
Comments to the Authors
The manuscript entitled "Mechanistic Insights into Salvigenin for Glucocorticoid-Induced Femoral Head Osteonecrosis: A Network Pharmacology and Experimental Study" is focused on an important topic, especially considering the effects of Salvigenin demonstrated, which hold significant potential for treating Glucocorticoid-Induced Femoral Head Osteonecrosis. The manuscript addresses a highly pertinent area of research and could provide valuable insights into future therapeutic approaches. However, several issues have emerged that prevent me from recommending the manuscript for publication in its current form.
1. The document should be reviewed for the presence of repeated definitions of abbreviations (e.g., ONFH in lines 90, 100, 200, etc.). For the convenience of the reader, it is recommended that abbreviations be defined only on their initial mention in the text, with a table providing a comprehensive list of abbreviations and their definitions.
2. The document should be checked for the presence of full stops before the reference at the end of sentences (e.g., as in lines 38, 47, 62, etc.).
3. Since the CCK8 analysis is highly variable, it is essential to specify the number of repeats measured in each sample (Figure 4, A-C). This should also be reflected in the demonstrated formula. This applies to all experimental data in figure 4 (F-K), figure 5 (E-F).
4. The illustrations throughout the document are challenging to discern (especially figures 1-3), with some proving unreadable. It is imperative to meticulously review all figures within the article, make text in the figures bigger and ensure their clarity to the reader.
5. In figure 4E, the images of the blots require alignment. Additionally, please elucidate the nature of the graphical artefact observed in the top right corner of the ESR1 bands image.
6. The value of 'n' for the experiments should be specified, e.g. what is the value for Figure 4? In the picture of the blots, only two bands in each group are demonstrated, whereas a minimal statistically significant n=3. If data from different membranes was analyzed, the statistical methods used in these calculations should be detailed, as well as the reason for not using (if any) 'interblot control'.
7. The pictures of GAPDH in Figures 4E and 5D appear strikingly similar. Please clarify whether these are the same GAPDH bands or different ones. Additionally, in Figure 5D, it would be preferable to ensure that the GAPDH bands align precisely with the OPN and RUNX2 protein binds, otherwise there is a risk of the GAPDH image appearing as if it was captured on a different membrane.
8. In Figure 6B, the alterations in mitochondrial morphology are challenging to discern. Therefore, please provide images at higher magnification and, if feasible, a quantitative analysis of the mitochondrial network parameters in the cells. Please indicate which mitochondria (green) and mitochondrial markers (red) you have stained (what special dyes or antibodies you used). In the case that it is not possible to quantify changes in mitochondrial morphology, it is imperative to indicate in this block and «Discussion» that such a conclusion was made by the authors based purely on their subjective observations.
9. The Discussion section could benefit from enhanced self-criticism. For instance, one of the primary questions remains unaddressed: how does Salvigenin affect the therapeutic activity of glucocorticoids?
10. Linae 401: What is GIOFN?

Author Response
Dear reviewer,
We are profoundly grateful to the reviewer for your comprehensive and meticulous review. Your deep knowledge and thoughtful critiques have played an essential role in refining our manuscript, allowing us to improve both its scientific clarity and methodological robustness. We sincerely appreciate the time and effort you have devoted to this process, as well as your commitment to advancing research in this important field. Your expertise and guidance have been invaluable, and we greatly respect your professional contributions. In response to your specific comments, we have carefully revised the manuscript as detailed below.
Comments 1:
The document should be reviewed for the presence of repeated definitions of abbreviations (e.g., ONFH in lines 90, 100, 200, etc.). For the convenience of the reader, it is recommended that abbreviations be defined only on their initial mention in the text, with a table providing a comprehensive list of abbreviations and their definitions.
Response 1:
Thank you for this valuable suggestion. We acknowledge that certain abbreviations, such as ONFH (Osteonecrosis of the Femoral Head), have been redefined multiple times throughout the manuscript. To enhance readability and maintain consistency:
- We have ensured that each abbreviation is defined only at its first occurrence in the main text.
- We have removed redundant definitions throughout the manuscript.
- We have included a table listing all abbreviations and their full forms in the Abbreviations section at the end of the manuscript for the convenience of the reader.
We appreciate the reviewer’s insightful feedback, which has helped improve the clarity and readability of the manuscript.
Comments 2:
The document should be checked for the presence of full stops before the reference at the end of sentences (e.g., as in lines 38, 47, 62, etc.).
Response 2:
Thank you for your careful review of our manuscript. We appreciate your attention to detail regarding punctuation consistency. We have now systematically checked and corrected all occurrences where full stops were incorrectly placed before the reference citations.
We appreciate the reviewer’s feedback, which has helped improve the manuscript’s formatting and readability.
Comments 3:
Since the CCK8 analysis is highly variable, it is essential to specify the number of repeats measured in each sample (Figure 4, A-C). This should also be reflected in the demonstrated formula. This applies to all experimental data in Figure 4 (F-K), Figure 5 (E-F).
Response 3:
Thank you for your valuable suggestion. We appreciate the importance of specifying the number of experimental repeats to enhance clarity and reproducibility. In response to this comment, we have added the relevant information regarding the number of replicates in the figure legends in the revised manuscript.
We appreciate this insightful feedback, which has helped improve the transparency of our data presentation.
Comments 4:
The illustrations throughout the document are challenging to discern (especially Figures 1–3), with some proving unreadable. It is imperative to meticulously review all figures within the article, make text in the figures bigger, and ensure their clarity to the reader.
Response 4:
Thank you for your valuable feedback. All figures in the manuscript were created using Adobe Illustrator as vector graphics, ensuring high resolution and scalability. When viewed at their original size or enlarged, the illustrations remain clear and fully legible.
To further address this concern, we have uploaded the original full-size figures alongside the revised manuscript for reference. Additionally, we have ensured that all key findings are clearly described within both the figure legends and the results section, allowing readers to interpret the data effectively even without relying solely on the figures. Given these considerations, we believe that the clarity of our figures does not compromise the readability or understanding of the manuscript.
Nonetheless, we appreciate the reviewer’s suggestion and have carefully reviewed the figures to further optimize text readability where necessary.
Comments 5:
In Figure 4E, the images of the blots require alignment. Additionally, please elucidate the nature of the graphical artefact observed in the top right corner of the ESR1 bands image.
Response 5:
Thank you for your careful review of our figures. We have realigned the blot images in Figure 4E to ensure proper visual consistency. Regarding the graphical artefact in the top right corner of the ESR1 bands image, we have identified it as a residual marker stain from the membrane labeling process. We appreciate the reviewer’s attention to detail, which has helped improve the clarity and presentation of our figures.
Comments 6:
The value of 'n' for the experiments should be specified, e.g. what is the value for Figure 4? In the picture of the blots, only two bands in each group are demonstrated, whereas a minimal statistically significant n=3. If data from different membranes was analyzed, the statistical methods used in these calculations should be detailed, as well as the reason for not using (if any) 'interblot control'.
Response 6:
Thank you for your insightful comment. We recognize the importance of clearly specifying the sample size (n) and ensuring methodological transparency in our Western blot analysis.
To clarify, in our Western blot experiments, we used a 15-lane gel and adopted a loading strategy designed to minimize reagent waste while maintaining statistical robustness. Specifically, we ran another set of samples on the other side of the same gel, ensuring that each group was run in quadruplicate on a single membrane. The first, eighth, and fifteenth lanes were used for markers, while protein samples were loaded into the intermediate lanes. Thus, although only two representative bands per group are displayed in Figure 4E, the quantification was based on at least three independent biological replicates (n = 3).
Additionally, we have ensured that all Western blot data were normalized using internal loading controls (GAPDH) to account for potential variability across lanes. Since all replicates were analyzed on the same membrane, interblot control was not necessary. However, to enhance clarity, we have provided further details on our Western blot methodology, including the statistical approach used for quantification, in the Methods section of the revised manuscript.
These modifications have been incorporated into the manuscript (line 267-284), and we appreciate the reviewer’s comment, which has helped improve the clarity and rigor of our experimental design.
Comments 7:
The pictures of GAPDH in Figures 4E and 5D appear strikingly similar. Please clarify whether these are the same GAPDH bands or different ones. Additionally, in Figure 5D, it would be preferable to ensure that the GAPDH bands align precisely with the OPN and RUNX2 protein bands; otherwise, there is a risk of the GAPDH image appearing as if it was captured on a different membrane.
Response 7:
Thank you for your insightful comment. We fully understand the importance of ensuring transparency in Western blot presentation.
We would like to clarify that the GAPDH bands in Figures 4E and 5D originate from different loading wells. Since GAPDH is a well-established housekeeping protein, its expression is highly stable under identical experimental conditions, and thus, when scanned from the same batch of protein samples, the band patterns may appear similar. However, a detailed examination of grayscale intensity values and visual comparison confirms that these GAPDH bands are indeed different. Therefore, we assure the reviewer that no bands have been reused between figures.
Regarding the alignment issue of protein bands in Figure 5D, we acknowledge that the resizing and formatting adjustments for figure placement may have led to minor misalignment. To address this, we have carefully realigned the GAPDH bands with the corresponding OPN and RUNX2 bands in the revised figure to ensure visual consistency.
These improvements have been incorporated into the revised manuscript. We appreciate the reviewer’s attention to this detail, which has helped enhance the clarity and accuracy of our figure presentation.
Comments 8:
In Figure 6B, the alterations in mitochondrial morphology are challenging to discern. Therefore, please provide images at higher magnification and, if feasible, a quantitative analysis of the mitochondrial network parameters in the cells. Please indicate which mitochondria (green) and mitochondrial markers (red) you have stained (what special dyes or antibodies you used). In the case that it is not possible to quantify changes in mitochondrial morphology, it is imperative to indicate in this block and «Discussion» that such a conclusion was made by the authors based purely on their subjective observations.
Response 8:
Thank you for your insightful comment. We acknowledge the importance of clearly presenting mitochondrial alterations.
In this study, we used JC-1 staining, which is a well-established method for assessing mitochondrial membrane potential (ΔΨm). The principle of JC-1 staining is based on its aggregation state within mitochondria, rather than direct visualization of mitochondrial morphology. Specifically:
- In healthy, polarized mitochondria, JC-1 forms J-aggregates, emitting red fluorescence (~590 nm).
- In depolarized mitochondria, JC-1 remains in its monomeric form, emitting green fluorescence (~530 nm).
- This red-to-green fluorescence shift reflects functional mitochondrial status, rather than providing structural resolution of mitochondrial morphology.
Given that JC-1 is not designed for direct mitochondrial morphology analysis, we have clarified in both the Figure 6B legend and the Discussion section that our interpretation of mitochondrial changes is based on fluorescence intensity shifts rather than structural alterations.
Additionally, in Figure 6B, the green fluorescence represents JC-1 monomers in depolarized mitochondria, while the red fluorescence represents JC-1 aggregates in polarized mitochondria. If a direct morphological assessment were required, MitoTracker staining or TOM20 immunofluorescence would be a more suitable approach.
We appreciate the reviewer’s feedback, which has helped improve the clarity of our data presentation and methodology.
Comments 9:
The Discussion section could benefit from enhanced self-criticism. For instance, one of the primary questions remains unaddressed: how does Salvigenin affect the therapeutic activity of glucocorticoids?
Response 9:
Thank you for this insightful comment. We recognize the importance of discussing the potential impact of Salvigenin on the therapeutic activity of glucocorticoids (GCs).
In this study, we primarily focused on the protective effects of Salvigenin against GC-induced osteoblast dysfunction and mitochondrial damage, without assessing its potential impact on the broader therapeutic functions of GCs, such as their anti-inflammatory and immunosuppressive effects. This remains an important question for future investigation.
Currently, our data do not provide direct evidence that Salvigenin interferes with the systemic actions of GCs. However, given that GCs exert their effects largely through glucocorticoid receptor (GR)-mediated transcriptional regulation, future studies should investigate whether Salvigenin modulates GR signaling pathways in other tissues. If Salvigenin selectively protects osteoblasts from GC-induced damage without compromising the therapeutic benefits of GCs in inflammatory diseases, it would hold significant promise as an adjunctive treatment for patients receiving long-term GC therapy.
To reflect this, we have revised the Discussion section to explicitly acknowledge this question and highlight the need for further research (line 685-691 and line 716-721 in the revised manuscript). We appreciate the reviewer’s suggestion, which has strengthened the critical analysis of our findings.
Comments 10:
Line 401: What is GIOFN?
Response 10:
Thank you for pointing this out. This was a typographical error, and "GIOFN" should have been "GIOFH" (Glucocorticoid-induced osteonecrosis of the femoral head). We have corrected this mistake in the revised manuscript.We appreciate the reviewer’s careful attention to detail, which has helped improve the accuracy of our text.

Round 2
Reviewer 3 Report
Comments and Suggestions for Authors
The reviewer notes that the authors have done a great job in improving the quality of the manuscript and have addressed almost all of the comments.
To complete the work, the comments on Figure 6B need to be revised.
As is clear from the authors' response to the reviewer's comment on Figure 6B, we are talking about the staining of mitochondria with JC-1 dye: normal, polarized mitochondria are stained red, whereas depolarized mitochondria are stained green. It is this (rather than morphological integrity, which cannot be assessed in this case) that is reflected in Figure 6B, and the authors should comment on this in the text and legend to Figure 6B.

Author Response
Dear Reviewer,
We sincerely appreciate your time and effort in reviewing our revised manuscript. Your insightful comments and constructive suggestions have been invaluable in further refining our work. We are grateful for your positive feedback on the improvements we have made and for your keen observations that have helped us enhance the clarity and accuracy of our findings. Your expertise and thoughtful review process have significantly contributed to strengthening our manuscript.
Comments 1:
Figure 6B. As is clear from the authors' response to the reviewer's comment on Figure 6B, we are talking about the staining of mitochondria with JC-1 dye: normal, polarized mitochondria are stained red, whereas depolarized mitochondria are stained green. It is this (rather than morphological integrity, which cannot be assessed in this case) that is reflected in Figure 6B, and the authors should comment on this in the text and legend to Figure 6B.
Response 1:
Thank you for your insightful comment. We agree with your assessment and acknowledge that Figure 6B primarily reflects mitochondrial membrane potential rather than morphological integrity. To address this, we have revised the corresponding text in the Results section to explicitly describe JC-1 staining as a method for assessing mitochondrial membrane potential. Specifically, we now clarify that normal, polarized mitochondria exhibit red fluorescence, whereas depolarized mitochondria show green fluorescence, and that dexamethasone treatment induces mitochondrial depolarization, which is partially restored by Salvigenin co-treatment.
Furthermore, we have updated the legend of Figure 6B to reflect this clarification. These revisions can be found in lines 587–595, 600-605, and are marked in red in the revised manuscript.
